# Integrating AI/ML Models for Patient Stratification Leveraging Omics Dataset and Clinical Biomarkers from COVID-19 Patients: A Promising Approach to Personalized Medicine

**DOI:** 10.3390/ijms24076250

**Published:** 2023-03-26

**Authors:** Babatunde Bello, Yogesh N. Bundey, Roshan Bhave, Maksim Khotimchenko, Szczepan W. Baran, Kaushik Chakravarty, Jyotika Varshney

**Affiliations:** VeriSIM Life Inc., 1 Sansome Street, Suite 3500, San Francisco, CA 94104, USA

**Keywords:** SARS-CoV-2, COVID-19, artificial intelligence, omics, patient stratification, risk management

## Abstract

The COVID-19 pandemic has presented an unprecedented challenge to the healthcare system. Identifying the genomics and clinical biomarkers for effective patient stratification and management is critical to controlling the spread of the disease. Omics datasets provide a wealth of information that can aid in understanding the underlying molecular mechanisms of COVID-19 and identifying potential biomarkers for patient stratification. Artificial intelligence (AI) and machine learning (ML) algorithms have been increasingly used to analyze large-scale omics and clinical datasets for patient stratification. In this manuscript, we demonstrate the recent advances and predictive accuracies in AI- and ML-based patient stratification modeling linking omics and clinical biomarker datasets, focusing on COVID-19 patients. Our ML model not only demonstrates that clinical features are enough of an indicator of COVID-19 severity and survival, but also infers what clinical features are more impactful, which makes our approach a useful guide for clinicians for prioritization best-fit therapeutics for a given cohort of patients. Moreover, with weighted gene network analysis, we are able to provide insights into gene networks that have a significant association with COVID-19 severity and clinical features. Finally, we have demonstrated the importance of clinical biomarkers in identifying high-risk patients and predicting disease progression.

## 1. Introduction

Over the past two years, the global COVID-19 pandemic has highlighted the crucial role of accurate patient diagnostics in preventing an overload of healthcare resources. The pandemic has led to a significant increase in patient surges globally, resulting in varying degrees of respiratory illnesses and a rise in mortality rates [1]. The symptoms of COVID-19 induced by the varying strains of the pathogen SARS-CoV-2 are challenging to differentiate from other common respiratory infections in a large proportion of those infected. Moreover, disease progression varies from asymptomatic cases to critical patient conditions, making it challenging for appropriate treatment selections and accurate prognoses. Some patients with COVID-19 rapidly develop severe dysfunctions and even critical illness, demonstrating an excellent example of infection variability due to internal physiological factors [2]. Various SARS-CoV-2 variants led to more concerns due to lowered vaccine efficacy and rapidly decreasing immunity. On the other hand, waves and variants of SARS-CoV-2 helped us understand the causes and effects of the COVID-19 catastrophe that was related to respiratory illness and multiorgan failure. The emergence of novel omicron hybrid variants such as BA (x), XE, XD, XF, etc., demonstrated the role of population variability in disease severity and outcome [3]. Furthermore, even after recovery, a portion of patients keep experiencing a spectrum of COVID-19-like symptoms generally termed “Long COVID”, making disease outcome prognosis even more challenging. Subsequently, this situation is substantially aggravated by the lack of understanding of the main signaling and pathogenetic mechanisms induced by COVID-19 infection in the host.

Correct risk evaluation and management in the realm of infectious diseases are critical for the diagnostic assessment and selection of the best line of therapies, markedly increasing the likelihood of patient survival [4]. The COVID-19 outbreak has highlighted the significance of implementing these measures. Therefore, one of the greatest challenges during the pandemic, especially in resource-strained settings, has been the early identification of individual patients at higher risk for adverse outcomes. Hence, it is critical to develop and implement intelligent risk-assessment tools that can predict a patient’s disease progression and recovery and suggest best-fit therapeutics for markedly reducing disease severity.

Data-driven risk evaluation based on artificial intelligence/machine learning (AI/ML) software models is considered an effective decision-making algorithm in the streamlining triage of emergency patients [5]. Sorting patients into different cohorts based on the disease severity and prognosis helps rationally allocate healthcare resources and identify the most effective therapy. AI-powered models have been recently used with success to simulate and predict drug–drug interactions in different patient groups, which aids in preventing potential issues during the initial stages of drug development in clinical trials [6]. Contemporary AI/ML-driven approaches in the healthcare space have accelerated the drug discovery and development pipelines due to the availability and prioritization of data alongside a wide range of statistical learning methods that leverage data to draw inferences and make new predictions [7]. AI technologies are able to recognize specific pathways responsible for the development of disease-related syndromes that leads to a more accurate categorization of the patient cohorts [8]. Additionally, computational assistance to healthcare providers contributes to reduced emotional pressure and, consequently, increased accuracy selection of successful therapeutic strategies [9,10].

Implementation of AI/ML-based patient stratification is a suitable strategy for developing custom patient-specific therapeutic guidance because of the sufficient availability of COVID-19 clinical data. Such strategies may contribute to overcoming the pandemic altogether as model applications move from being patient-based to population-based [5]. Current AI/ML algorithms providing COVID-19 prognosis belong to either image- or non-image-based categories. The first group significantly relies on chest X-ray or digital tomography readouts [11] that reflect lung function but cannot accurately quantitate the degree of immune response and upregulation of biomarkers in the inflammatory pathways. It was demonstrated that quantifying proinflammatory cytokine levels, blood, and urine biomarkers improves the assessments of patient clinical conditions, showing high accuracy [12]. Non-image-based methods for COVID-19 include using electronic health record (EHR) information to diagnose COVID-19 and assess its severity. These methods consist of two basic types of studies: score system-based and ML-based COVID-19 diagnostics [13]. Researchers seek to identify important predictors and assign them associated scores for the former. Subsequently, the summation of the scores can potentially lead to the stratification of patients with different disease severities [12,14]. Currently, several risk assessment scores are available to predict the severity of different, none-COVID-19-related diseases for ICU patients [15]. Furthermore, predictors indicating the need for intensive respiratory or vasopressor support for COVID-19 patients or suggesting the high mortality risk in COVID-19 patients with pneumonia have been identified [16,17].

Liang et al. [12] built a predictive risk score (COVID-GRAM) system, which included ten important predictive factors screened from 72 potential predictors among epidemiological, clinical, laboratory, and imaging variables. The COVID-GRAM was used to estimate the risk of developing critical illness for patients with COVID-19. Other methods are based on a gradient boosting decision tree model that predicts an individual’s COVID-19 infection progression, recovery rate, and mortality risk using demographic information such as age, sex, race, and a series of routine lab tests [18]. Recent advances in deep learning (DL) combined with EHR datasets have gained recent traction for diagnosing COVID-19. For example, a feedforward neural network-based DL survival model was used to predict critical illness development risk in COVID-19 patients by evaluating 74 baseline clinical features [12]. It has been noted that patient genomics data, which play a crucial role in the diversity of disease severity, are not accounted for in any of those models [19,20]. Adding genomics information into a predictive modeling system can help enrich and identify critical pathways and associated nuances contributing to the disease severity and select the most appropriate treatment options.

Understanding the biological pathways that impact severe cases of COVID-19 is crucial to identifying effective drug interventions and improving survival rates. The main objective of the study was to understand the most impactful biomarkers that contribute to severe cases and lower survival through ML predictive models (Figure 1). Models were trained on clinical patient data that include several biomarker levels in correspondence with case severity and survival. After achieving optimal performance, an analysis of model explainability was conducted to reveal the “black box effect” of the predictive models and identify the biomarkers and their corresponding values that have the greatest influence on model predictions. Upon identifying significant biomarkers, a correlation analysis was conducted between groups of genes and the biomarkers that were modulated to pinpoint the most probable cluster of genes contributing to severe cases of COVID-19 and decreased survival rates. ML modeling was employed to conduct a correlation analysis on a patient pool consisting of both OMICS and clinical datasets. Our AI/ML modeling has helped us reveal the significant association between the most influential gene cluster and the related biological pathways. This manuscript showcases our findings, which can greatly improve our comprehension of the features of potential drug candidates to address the severity and mortality of COVID-19.

## 2. Results

### 2.1. Model Outcomes

Performance metrics for various modeling architectures were collected in Appendix A (Appendix A). Missing values within the dataset was imputed using Multiple Iteration Chain Estimation (MICE) regression techniques [21], where each missing data point is iteratively learned from other available features. Boosted Decision Tree architectures, such as LightGBM, XGBoost, and CatBoost, have built in techniques for handling missing values [22], so evaluation metrics were reported twice (with or without MICE) for each of these architectures.

The balanced accuracy was 91.6%, and the ROC-AUC score was 98.1% for the best performing severity model. The balanced accuracy was 99.4%, and the ROC-AUC score was 99.9% with the best performing survival model. Both models were LightGBM classifier models, a lightweight modeling infrastructure that applies Gradient Boosting Machine (GBM) decision trees. GBMs consist of iterative, or “boosted”, decision trees that fit training feature thresholds to approximate predictions. The LightGBM model architecture incorporates missing value-handling logic that minimizes loss on the training dataset in the most effective way possible. The internal decision trees within the model itself represent missing values in such a way that it aligns most with the training data; however, it maintains the assumption that the value is unknown rather than inferring an actual value. The modeling logic’s robustness enables it to handle missing data as input, indicating that clinicians are not required to input all biomarker levels observed in the clinical training dataset columns to receive a valid prediction from the trained model. The high performing results indicate the usefulness of clinical features for both predictive use cases. The predictive modeling results are summarized in Table 1. ROC curves for severity and survival models are shown in Figure 2 and Figure 3, respectively.

The fact that the clinical features used to train both models can be generalized, along with the absence of some portions of the biomarker laboratory datasets, reinforces the practicality of deploying predictive model in a clinical setting. The input to the predictive models need not include all the listed blood biomarker values in the feature set to generate a dependable prediction, but rather only some of them.

In the next use case, training data was also modified only to include patients with no comorbidities to assess if similar biomarker-based features for non-comorbidity patients are representative of predicting COVID-19 case severity and survival for a cohort containing both comorbidity and non-comorbidity patients within the same test set. With this data filtering, the training set was reduced by 63%, making it reasonable for the evaluation metrics to drop. The results of this training data modification are shown in Table 2. ROC curves for severity and survival models are shown in Figure 4 and Figure 5, respectively.

Despite using a modified training set, the model performance results still demonstrate a robust test ROC-AUC score of 93.5% for predicting the severity of COVID-19. This suggests that the features without comorbidities remain highly indicative of COVID-19 case severity, regardless of whether the patients have comorbidities or not. However, the ROC-AUC score for the COVID-19 survival models is significantly lower with the training set adjustment, which indicates that certain comorbidities are strongly correlated with survival. These results suggest that it is unlikely to observe a strong influence of comorbidity features through the model explainability analysis for the COVID-19 severity model. However, impactful comorbidity features such as coronary artery disease and diabetes from model explainability analysis for the COVID-19 survival model can be noted during analysis.

### 2.2. Patient Biomarker Analysis

#### 2.2.1. Clinical Biomarker Evaluation

Model explainability has transformed the way ML can be applied, especially in the biomedical domain. Through model explainability methods, inferences can be gathered on whether higher or lower feature values contribute to a higher probability of a given predictive class. As high performance for both COVID-19 severity and survival models were achieved, model explainability analysis allowed for the evaluation of the biomarker values that are highly correlated with more severe COVID-19 and least survival of COVID-19 cases. Model explainability was achieved through SHAP (SHapley Additive exPlanations) analysis, which is an approach to explain the output of any machine learning model based on game theory that infers model interpretation of specific feature values based on the impact that they have on the model itself [23]. Specific feature values are transformed into *SHAP* values to measure the impact on the model prediction. Equation (1) below shows the calculation of the *SHAP* value for the nth feature among *N* feature subsets, given a feature subset without the nth feature (*S*), the total number of features (*F*), and the predictive model (*M*):(1)$$SHAP_{n,M}=\sum_{S\subseteq N\backslash \left\{n\right\}}^{}\frac{S|!\left(F-length\left(S\right)-1\right)!\;}{F!}\left[M\left(S\cup \left\{n\right\}\right)-M\left(S\right)\right]$$

The feature impact scores interpolated from the *SHAP* values for the top 20 most impactful features of the COVID-19 severity and survival predictive models are shown in Figure 6 and Figure 7, where higher feature impact scores indicated whether higher or lower feature values were correlated with more severe COVID-19 cases and lower chance of survival from COVID-19. The top 20 features were determined by the highest absolute SHAP value mean, which shows the most impactful features contributing to either class (severe case/moderate case) in Figure 6 (survival/mortality) and in Figure 7.

As implied by the results presented earlier, there were no significant comorbidity-based features in the COVID-19 severity model. However, it was interesting to note that the presence of comorbidities, coronary artery disease, and diabetes were strongly correlated with low survival. Moreover, there has been evidence to suggest the presence of coronary artery disease [24] and diabetes [25] as a predictor for both COVID-19 severity and mortality. It is also interesting to note that patients who had been given methylprednisolone had a higher probability of having a severe case and lower survivability. Although there is no established link between the identified medication and COVID-19 severity or patient survival in the clinical dataset, it would be valuable to examine the medication’s impact when used in conjunction with insights from impactful biomarkers. Furthermore, this is particularly relevant given that steroid anti-inflammatory drugs are frequently given to patients with severe and critical respiratory infections [26]. It is highly possible that severe patients were given the medication as a potentially preventive measure; however, from the analysis there is no indication of any positive or negative effect on their survival or severity.

Beyond comorbidities and medications, the following features, included biomarkers, have a specific pattern associated with severe COVID-19 cases and lower survival predicted by our AI/ML driven platform were further validated by external studies:Sequential Organ Failure Assessment (SOFA) Score: Higher SOFA score values are associated with more severe cases and lower survival [27].Lactate Dehydrogenase (LDH): Higher presence of LDH is associated with more severe cases and lower survival [28].Blood Urea Nitrogen (BUN): Higher presence of BUN is associated with more severe cases and lower survival especially with lower Serum Creatinine levels [29] and lower Albumin levels [30].

The following graphs shown in Figure 8 outline the specific range of values within scatter plots for impactful features as they pertain to COVID-19 severity.

The ranges of values were calculated by measuring the 5th and 95th quantile of biomarker values when the severity was high. The same logic was applied to survival models. The following graphs in Figure 9 outline the specific range of values within scatter plots for impactful features concerning COVID-19 survival.

The scatter plots indicated the clear correlation between certain biomarker ranges and feature impact on each respective model. These validated features were selected for the gene correlation analysis to see what specific group of genes were associated with the upregulation of the given features.

With the generated SHAP values (feature impact scores) for each given feature value, a clustering analysis was performed to see how patients were grouped with varying biomarker levels. The clustering analysis was performed on feature impact scores based on the COVID-19 severity model explainability analysis. The elbow method [31] was used to determine the optimal number of clusters, and K-Means clustering was the clustering modality that generated the final clusters. Observations for each cluster and biomarker box plots are shown in (Appendix A, Appendix A). It is worth noting that the distinction between the clusters were due to varying levels of the body weight, SOFA score, BUN, serum creatinine, LDH, albumin, PCO2 arterial, potassium levels, and lymphocyte count percent. Despite not showing any significant biomarker variation compared to the norm, Cluster 5 had a high percentage of severe COVID-19 cases among the patient group, with 93.1% affected. This suggests several less impactful biomarkers may have influenced the cluster’s formation.

#### 2.2.2. Omics Data Analysis of COVID19 Patients

Several gene co-expression modules were identified through gene network analysis, with MEcyan and MEdarkred standing out for their strong correlation with several key traits. Of these two modules, MEcyan was particularly noteworthy, as it showed a significant positive correlation with COVID-19 severity, end of organ damage (EOD), clinical biomarkers such as BUN, serum creatine, D-Dimer, creatine kinase, LDH, and SOFA score evaluation of patients. This finding aligns with the ML prediction referenced in Figure 10. However, only MEdarked indicates a moderately significant association with comorbidities such as chronic kidney disease (CRD), heart failure, and alcohol/non-alcohol liver disease comorbidity in patients, as shown in Figure 11. In this study, MEcyan and MEdarkred gene modules were selected for functional annotation analysis and also analyzed for biological significance in the pathways across patients’ traits, COVID-19 severity, patient disease comorbidity, and major clinical biomarker record of the patients.

The genes present in the MEcyan module were subjected to functional enrichment analysis, which revealed their enrichment in pathways related to inflammatory and diverse immune responses, as per the Gene Ontology (GO) enrichment annotation (Figure 12). Additionally, they were found to be enriched in signaling pathways such as Tumor Necrosis Factor (TNF) signaling, cytokine signaling, Toll-like receptor signaling, and Interleukin-17 (IL-17) signaling pathway, based on the KEGG pathway (Figure 13). Additionally, several investigations have also confirmed our AI/ML-based predictive outcomes, demonstrating that these gene clusters were closely associated with immune signaling and response pathways that help combat viral infections [32,33,34]. The MEdarkred module’s overrepresented pathways included platelet activation, regulation and response to blood coagulation, wound healing pathway, cell motility, and hemostasis for its genes. Those were considered important as the gene module results showed negative correlations with heart failure, chronic artery disease, and liver disease in patients (Figure 11 and Figure 14).

The estimated GS and MM genes are indications of how biologically significant such genes are in the enriched pathways in relation to the traits of interest (Appendix A). Our findings suggest that MEcyan modules harbor genes that exhibit a favorable association with COVID-19 severity markers, such as EOD, SOFA score, LDH, and BUN, while showing an unfavorable association with comorbidities such as heart failure and kidney disease (Appendix A). MEdarkred modules show genes with a negative correlation with EOD, BUN, and SOFA, but a positive correlation with LDH. By applying our AI/ML modeling, based on both high significance and high intramodular connectivity, several noteworthy biomarkers were identified in the MEcyan and MEdarkred modules that are associated with traits such as EOD, SOFA, LDH, with GS > 0.3, and MM > 0.8. These gene-biomarkers (Appendix A) could be investigated in more detail for their potential use as drug targets or diagnostic tools. In addition, the analysis of patient OMICS datasets using AI/ML modeling has identified specific genes, including P2Y12, ECE1, MSANTD3-TMEFFI, PLEKHA8P1, NUTF2, SAV1, CXCR2P1, and MSANTD3, within the MEdarkred gene module that exhibit high GS and MM for clinical biomarkers such as SOFA score, BUN, EOD, and serum creatine. These genes are involved in biological pathways associated with blood coagulation and wound healing and have been implicated in the regulation of clinical biomarkers in patients with COVID-19.

## 3. Discussion

We have developed a robust AI/ML-based model that links OMICS datasets to clinical biomarkers for the stratification of patients and accurate predictions of disease severity and survival in a given cohort of COVID-19 patients. Given the high ROC-AUC score for both models, it can be inferred that the clinical features were sufficient to be an indicator for both COVID-19 severity and survival. However, the impact of each feature on the disease development and severity remains unclear. ML modeling was proposed as one of the promising approaches to identify the role of biomarker pathways in infection disease development. Certain restrictions on the dataset brought up potential model uncertainties as well. Sparsity in lab biomarker data restricts complete multivariate analysis for model explainability, as there would be a possibility for even more granular feature observations if lab biomarker data were more complete. There is a possibility that certain biomarkers, such as Hemoglobin A1C and Interleukin 8, which had more than 90% missing values, may have contributed to a significant impact on model predictability. The lack of diversity among observed comorbidities within patients in the dataset may have limited the potential observation for certain comorbidities to contribute to COVID-19 severity and survival based on model conclusions.

ML has been recognized as an insightful utility to predict the severity and survival of patients with COVID-19. A study published by Raman et al. in 2023 [35] highlighted the use of ML models to predict COVID-19 severity among 1795 patients from the University of Texas Southwest Medical Center. The study showed high predictive performance (ROC-AUC = 0.81, 0.82) among the trained models and highlighted what the trained models ranked as the most impactful biomarkers. There was no indication of biomarker data sparsity in the dataset used in the study. The models within the current analysis are much more dynamic as the complete set of biomarker features are not required per patient when predicting COVID-19 disease severity or survival. Additionally, the application of SHAP within the current analysis allowed for the exploration of another dimension within the features, which gave insights to specific feature value impact on model performance rather than indicating that given features may be important. In a study published by Ikemura et al., SHAP was employed in ML models that predicted COVID-19 mortality, which also achieved high model predictive performance (ROC-AUC = 0.803) [36]. However, the dataset within the study contained only 48 patient variables compared to 135 within the current analysis—leading to a larger variety of feature insights. The high model performance as well as considerations for missing data and the abundance of biomarker data makes the ML approach for current analysis highly dynamic and scalable, especially in a real-world scenario where patient data can be integrated to generate predictions.

SHAP was used to determine the most impactful clinical features and feature value ranges contributing highly to COVID-19 severity and survival. To determine the significant biomarkers for each use case, the top 20 most influential features were identified. It is noteworthy that age, body weight, and BMI were among the top 20 impactful features for both cases, indicating that clinicians should prioritize patients falling within the high impact ranges for these categories. Comorbidities for coronary artery disease and diabetes were seen within the top 20 impactful clinical features (conditions) for survival. However, comorbidities did not seem to have a significant impact when predicting COVID-19 severity, as they were not in the top 20 factors influencing disease severity. It is possible that comorbidities could exacerbate COVID-19, resulting in fatalities even in cases where the severity of the illness is not high. Surprisingly, the presence of methylprednisolone as a medication for patients with COVID-19 was seen as an impactful clinical feature for both severity and survival. There is no conclusive evidence to suggest that the medication itself increased severity or reduced survival. Subsequently, this may suggest the low efficiency of steroid therapy in COVID-19 patients. Therefore, it is probable that the medication’s effect on certain biological pathways differs in different patient cohorts, which may influence significant biomarkers modulations found among this investigation. Another group of clinical features present in the top 20 impactful features for both models were blood biomarkers, namely BUN, LDH, serum creatinine, and albumin. In addition, an SOFA score indicating a clinical organ damage measure was seen as a highly impactful feature for both use cases as well. The clinical biomarkers and SOFA score impact was aligned to the gene expression analysis, where highly correlated gene groups were identified in the dysregulation of clinical biomarkers in COVID 19 patients.

The most impactful SOFA score range for COVID-19 severity and mortality was 4–14 and 3–14, respectively. These ranges tie closely with observations seen in [27], which suggests that patients with SOFA scores greater than or equal to 5 have a higher risk of mortality. In regard to LDH [28] reports, patients with severe COVID-19 have greater than 350 U/L for their mean concentration, which is slightly less than the lower value of the most impactful median LDH concentration range of 404 and 439 for severity and mortality, respectively. A larger ratio between BUN and creatinine levels has shown to be an indicator for COVID-19 severity with optimal thresholds at 33.5 and 51.7 according to [29]. The most impactful range for both BUN and serum creatinine, according to SHAP analysis, suggests that the most impactful BUN/creatinine ratio is between 21 and 355, which is a larger range window than the study suggests. This wide range is likely due to the sparsity of biomarker data and the independent contributions of each BUN and serum creatinine to the model predictions; however, the higher impact on severity and survival seen for higher BUN and lower serum creatinine values does suggest that a larger ratio would be an indicator for both use cases. Similarly, a larger ratio between BUN and albumin levels has shown to be an indicator for COVID-19 severity with optimal thresholds greater than 3.9 according to [30], and SHAP analysis suggests that the most impactful range of BUN/albumin is between 7.5 and 93. The literature validates the usefulness of the BUN/creatinine and BUN/albumin ratios. However, to enhance the value further, it would be beneficial to include additional features in the data. This could involve calculating the actual BUN to serum creatinine and BUN to albumin ratios, along with incorporating more laboratory biomarker data.

We conducted a weight co-expression gene network analysis in addition to our ML prediction. This allowed us to approximate the patient gene expression and pinpoint the genes that display a noteworthy correlation with COVID-19 severity, disease comorbidity, and the most critical clinical biomarkers. Our predictive analysis results suggest that there is functional enrichment in pathways related to inflammation, cytokine response, and Toll-like and interleukin signaling in the genes network that are correlated positively with COVID-19 patients having high BUN, LDH, serum creatinine levels, an elevated SOFA score, and EOD. Within the MEcyan modules, several crucial genes display characteristics such as interleukin receptors (IL8RBP, IL10RB, IL17RA) and Toll-like receptors (TLR1, TLR5, TLR4, TLR6, TLR8, TLR2) alongside MYD88. These genes are intricately associated with activating pro-inflammatory cytokines, signaling pathways, and sensing the SARS-COV-2 envelope protein [34,37]. In addition, TLR2 and MYD88 have been shown to be associated with COVID-19 disease severity. It is important to acknowledge that cytokines, despite being integral to the innate immune response to viral infections, can cause significant harm to organs and tissues when dysregulated or overly inflamed, leading to the onset of cytokine storms [3,4,5,6,7,8,9,10,11,12,13,14,15,16,17,18,19,20,21,22,23,24,25,26,27,28,29,30,31,32,33,34]. Hence, patients with hyperinflammation are often placed on immunosuppression, immunomodulation, and selective cytokine blocking drugs to improve their outcome [32,38,39]. Additionally, interleukin 10 receptor, IL10RB, was identified to be significant to patients’ severity and has been previously noted as a regulator of host susceptibility to COVID-19 severity and potential drug target [40,41].

Furthermore, the set of significant genes represented in the MEdarkred module (Appendix A) across EOD, SOFA, BUN, and heart failure, alcohol liver disease, and kidney disease comorbidity include gene markers such as AMAD9, KIF1B, SLC22A4, MAPK14, MAP2K6, SLC2A3, ILI7RA, which have been associated with COVID-19 disease severity, tissue injury, and sepsis. For instance, an increased expression of ADAM9 and others member of the metalloprotease protein family in the blood has been associated with tissue damage and identified as a potential drug target for COVID-19 [42,43,44,45,46,47]. Rinchai et al. particularly highlight the involvement of AMAD9 metalloprotease in cellular adhesion, protein shedding, various cell population and activity including cell–cell interaction, and fibroblast activity—a key part of connective tissue and extracellular matrix (ECM) formation. This is consistent with our findings where the annotation of the MEdarkred gene module is shown to be enriched in cellular component (Appendix A) activities such as the cell surface, focal adhesion, cell cortex, anchoring junction, stress fiber, platelet activities, myofibrils, and actin cytoskeleton. These networks of biological and cellular activities further emphasize the importance of better understanding observed abnormalities in the extracellular matrix (ECM), inflammatory response, activities of metalloproteinase proteins, and other key significant gene biomarkers in COVID-19 patients.

Our AI/ML platform demonstrated that most disease comorbidities exhibited minimal correlation, except for alcohol/non-alcoholic liver disease, diabetes, chronic kidney disease, and heart failure, which showed moderately significant negative correlation in the co-expression MEdarkred gene modules. These genes were enriched in pathways related to wound healing, blood coagulation, hemostasis, and cell motility, as shown in Figure 11 and Figure 14. This biological insight strengthens our previous observations regarding the link between hyper-inflammation, cytokine reactions, cytokine storms, and tissue and organ damage. This understanding may clarify the moderate correlation observed in gene modules for chronic kidney disease, heart failure, and liver disease with functional annotation enriched in wound healing, hemostasis, blood clotting control, and coagulation. Identifying significant gene-clinical biomarkers with high GS and MM scores can facilitate the selection of key genes for potential drug targeting based on their repositioning and druggability in the context of personalized medicine.

## 4. Materials and Methods

### 4.1. Clinical Data Acquisition

An analytical review of the literature indexed in the PubMed/MEDLINE and Scopus databases was conducted. The goal was to summarize all available sources providing clinical and OMICs data for individual patients infected with SARS-CoV-2 and admitted to the healthcare institutions. To identify clinical publications relevant to COVID-19 patients with appropriate datasets, a search strategy was employed using both broad terms and specific biomarker terminology to maximize the number of identified publications. We included peer-reviewed, pre-proof, and papers published ahead of print that reported COVID-19 cases, confirmed using a real-time reverse transcriptase-polymerase chain reaction. Articles in English and non-English languages were selected. The first run of the literature search using general terminology returned more than three hundred thousand article titles. Following, the search ranking algorithm was applied to triage the sources found, and twenty-two research articles with the primary patient data (Appendix A) were evaluated and used as a database for further ML training.

Specific sets of clinical terms with assigned logical weight for biomarker identification were deployed in the AI algorithm to further rank the source according to their relevance in providing individual clinical data.

### 4.2. Data Curation

The clinical dataset consisted of patient conditions, lab test results, and clinician reports including a Sequential Organ Failure Assessment (SOFA) score, which is used to predict ICU mortality based on lab results and clinical data. A SOFA score is considered helpful for ML model training because it directly reflects the patient’s condition based on the oxygen saturation, respiratory function, platelets count, blood pressure values, creatinine, and bilirubin test values as well as the Glasgow coma scale. SOFA score has demonstrated its reliability during its use for more than 25 years since it was developed [48]. The full clinical datasets for these patients were downloaded from Synapse with the identification number syn35874390 from synapse.org. The dataset was filtered to include patients that had tested positive for COVID-19 through either a PCR and/or antibody tests. There were 581 unique patients with a combined set of 7707 clinical data points containing multiple days of patient information. Individual patient parameters were categorized into demographic parameters, comorbidities, blood cell count parameters, and biochemical and inflammatory biomarkers. Duplicates were dropped to result in the overall 7707 clinical data points to ensure that unique patient information would be used to train the machine learning model. The gene expression dataset for this analysis was obtained from the Gene Expression Omnibus (GEO) with the accession number GSE215865. The dataset consisted of the transcriptomics (RNA-seq) of whole blood samples of hospitalized COVID-19 patients and healthy, hospitalized control patients [49]. Using the metadata available, the gene expression was reduced to 1198 samples after removing sets of patients with largely incomplete clinical descriptions [50,51].

### 4.3. Bioinformatics Methodology

Gene network analysis was conducted on 1198 gene expression data with the WGCNA package [52] to identify significant correlation between co-expressed gene modules and patients’ disease severity, comorbidity, and clinical biomarkers. First, differentially expressed genes (DEGs) between COVID-19-positive and COVID-19-negative (control) patients were extracted using normalized gene counts (at *p*-value < 0.05 and abs(logFC) >= 1) using limma package [53], and fed into construction of weighted gene co-expression networks. Using WGCNA functions, approximate scale free topology was determined to define the adjacency matrix and transformed into topological overlaps (TOM) and dissimilarity (dissTOM) matrix. A hierarchical clustering was performed to identify gene modules with the minimum size set at 30 genes. Modules with gene expression similarity were subsequently merged based on module eigengene and cluster correlation. Additionally, to determine key genes associated with COVID-19 severity and other clinical features, Gene Significance (GS) and Module Membership was calculated for every gene in the modules and filtered at GS > 0.2 and MM > 0.8. Genes with high values typically have high connection and interaction, and do have strong associations with traits of interests [52]. The selected key gene biomarkers are used for further analysis for biological annotations and activities. To gain insights into the biological role of these genes, functional annotation analysis was performed using Gene Ontology (GO) [54,55] and Kyoto Encyclopedia of Genes and Genome (KEGG) [56] using ShuyGO web-based tools [57].

### 4.4. Descriptor Analysis and Selection

For the initial dataset used in model training, we collected various types of data such as patient condition, biomarkers, comorbidities, and therapy information. The data were classified as either numerical or categorical data. The dataset also had biomarkers with varying levels of missing values, each of which was identified in the missing values table (Appendix A). Each relevant column used for model training is indicated in the column groups table (Appendix A).

All features were then normalized and evaluated to discern relative importance to the respective target features (survival outcome and disease severity) using our ML infrastructure pipeline. These algorithms were optimized to the data available in the training set. Algorithm performance in selecting essential features were proportional to the training data size [58]. BIOiSIM™’s AI infrastructure included automated statistical learning algorithms prioritizing a large collection of features and selecting those features that improved model performance. This subset was further refined during model training through Recursive Feature Elimination (RFE) [59] and Boruta selection algorithms [60].

### 4.5. Model Training and Evaluation

We developed two different types of ML models for this project. Both classification models used the biomarkers information, comorbidities, and therapy information to predict either COVID-19 case severity or survival outcome.

Inbuilt MLs were used for both classification and regression models. The ML algorithms used in the study included linear algorithms such as the Bayesian ridge, support vector machines, tree-based ensemble methods such as LightBoost, CatBoost, and XGBoost, random forests, and multi-layer perceptions [61]. Features selected from the initially assembled descriptor set were used as inputs in the training process.

All individual patient data contained information regarding the survival outcomes and length of hospital stay as the initial set for model training and evaluation. They were considered critical for model learning. The ML infrastructure randomly divided the entire dataset into 80% and 20% portions for the training and test splits. Hyperparameter optimization was conducted on the validation sets obtained through 10-fold stratified cross-validation of the training set, which involved creating validation sets using stratified splits. Model evaluation metrics, available in Section 2, were determined for the final unseen test holdout set, which was distinct from the training and cross-fold validation sets used in model training.

Classification model evaluation metrics include balanced accuracy and a ROC-AUC score (Area Under the Receiver Operating Characteristic Curve). Balanced accuracy is defined as an optimal approach to the standard accuracy metric, which is adjusted to perform better on imbalanced datasets. The ROC-AUC score assesses the distinction of probability predictions between two classes such as severe COVID-19 or moderate in the context of severity predictions. Balanced accuracy was selected since the survival model training dataset was imbalanced and to maintain consistent evaluation with the severity models. The ROC-AUC evaluation enables for the assessment of the predicted probabilities, which could be advantageous if the models were to be utilized in a clinical environment. This approach provides probabilistic information regarding the status of a patient with COVID-19. Subsequently, both metrics were determined on each test set from each respective trained model.

## 5. Conclusions

A robust AI/ML-based model was created to stratify COVID-19 patients using OMICS, and clinical biomarker datasets, which enables accurate prediction of disease severity and outcomes. The accuracy of both models was 98.1% and 99.9%, respectively. The stratification module is based on the ML/DL algorithms with built-in classification and regression models. A unified dataset comprising of clinical biomarkers and the OMICS dataset were used as primary inputs, and their multi-dimensional unified analysis helped identify, with high accuracy, critical transcriptomic and clinical biomarkers for the prediction of disease severity and outcomes. Our AI/ML-driven platform also demonstrated that comorbidities such as alcohol/non-alcoholic liver disorders, diabetes, chronic kidney disease, and heart failure could impact the severity of the disease and the outcomes of COVID-19 patients. The genomic analysis of the patients as a whole revealed the pathways implicated in the development of the disease along with related clinical biomarkers. Our research has demonstrated that patient stratification models, driven by AI/ML modeling, can be expanded to other viral infections, and could be used to precisely identify the manifestation of clinical biomarkers, resulting in more accurate diagnoses and treatment options in the context of personalized medicine.

## Figures and Tables

**Figure 1 ijms-24-06250-f001:**
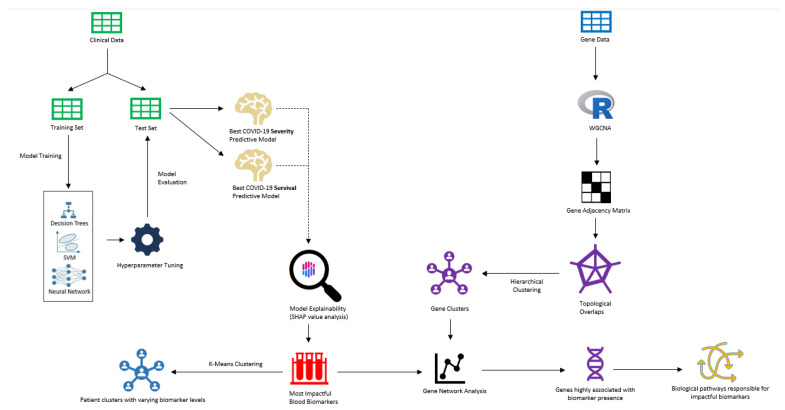
Overview of the workflow for an AI/ML-driven model that predicts the outcome of COVID-19 infection in patients, utilizing a set of clinical biomarkers and genomics dataset.

**Figure 2 ijms-24-06250-f002:**
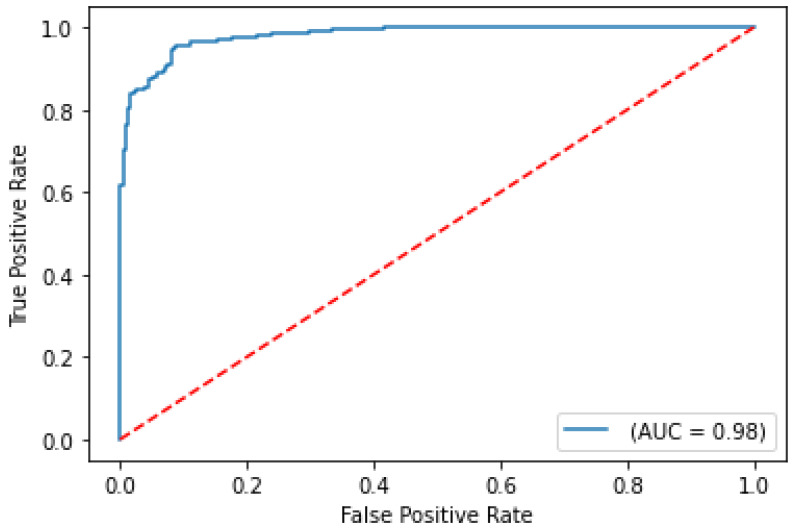
ROC curve on test set for COVID-19 severity prediction model.

**Figure 3 ijms-24-06250-f003:**
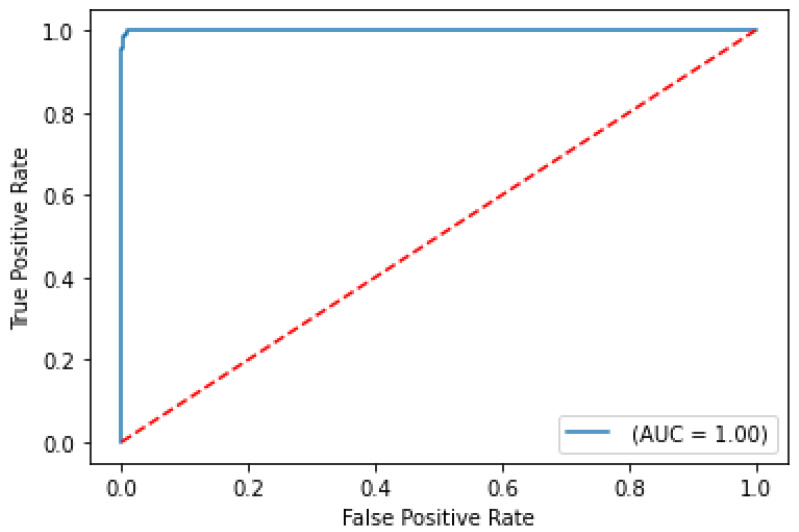
ROC curve on test set for COVID-19 survival prediction model.

**Figure 4 ijms-24-06250-f004:**
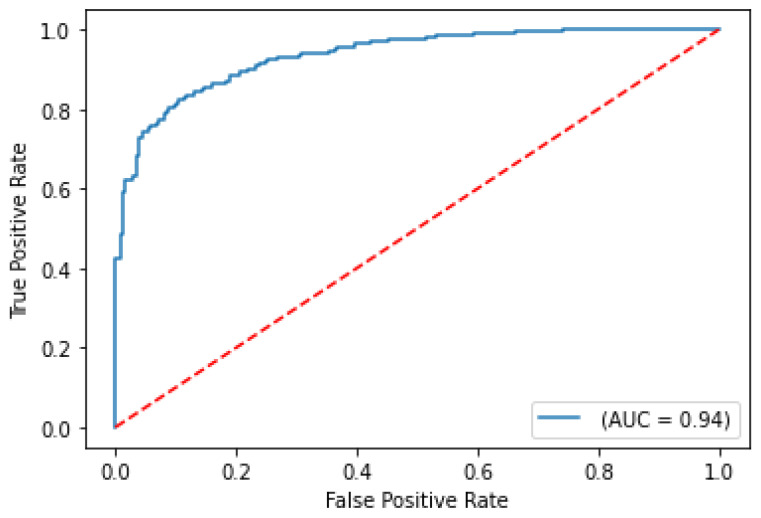
ROC curve on test set for COVID-19 severity prediction model with comorbidities re-moved from training set.

**Figure 5 ijms-24-06250-f005:**
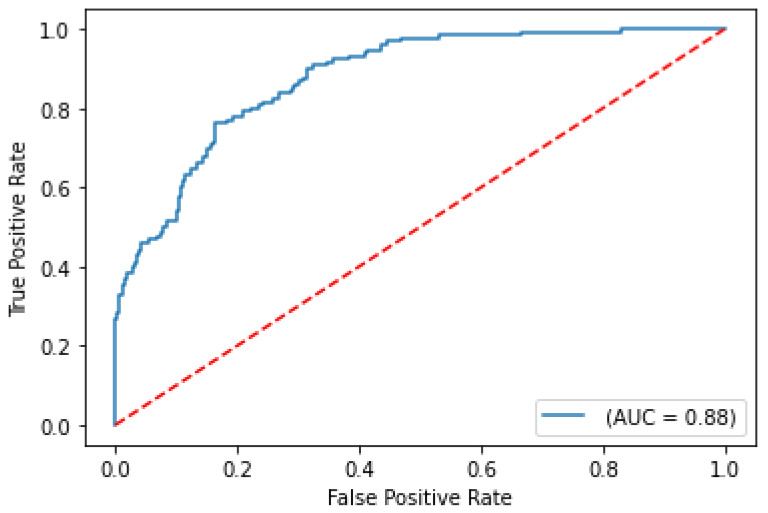
ROC curve on the test set for COVID-19 survival prediction model with comorbidities removed from the training set.

**Figure 6 ijms-24-06250-f006:**
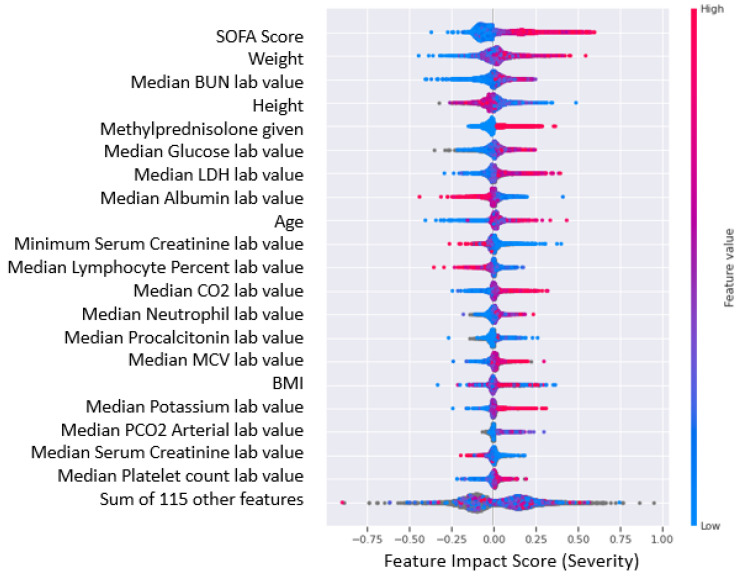
Top 20 most impactful features for COVID-19 severity predictive model. The red data points indicate higher values for the particular feature, while blue data points indicate lower values for the particular feature. A higher feature impact score indicates a higher contribution to a “severe” case prediction, while a lower feature impact score indicates a higher contribution to a “moderate” case prediction value.

**Figure 7 ijms-24-06250-f007:**
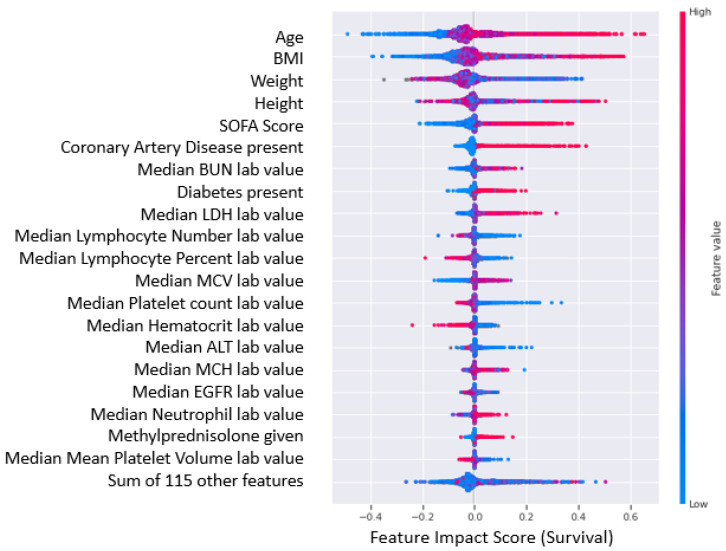
Top 20 most impactful features for COVID-19 survival predictive model. The red data points indicate higher values for the particular feature, while blue data points indicate lower values for the particular feature. A higher feature impact score indicates a higher contribution to a “no survival from COVID-19” prediction, while a lower feature impact score indicates a higher contribution to a “survival from COVID-19” prediction value.

**Figure 8 ijms-24-06250-f008:**
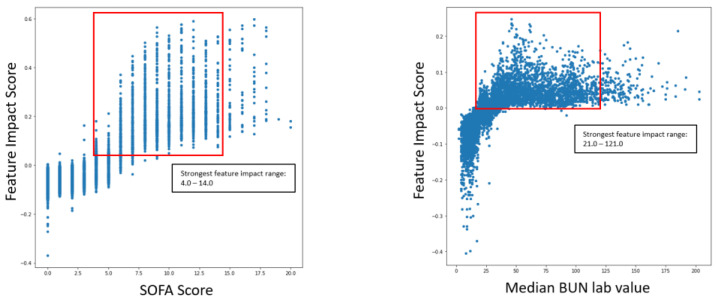
Range of impactful and validated biomarker values for COVID-19 severity prediction.

**Figure 9 ijms-24-06250-f009:**
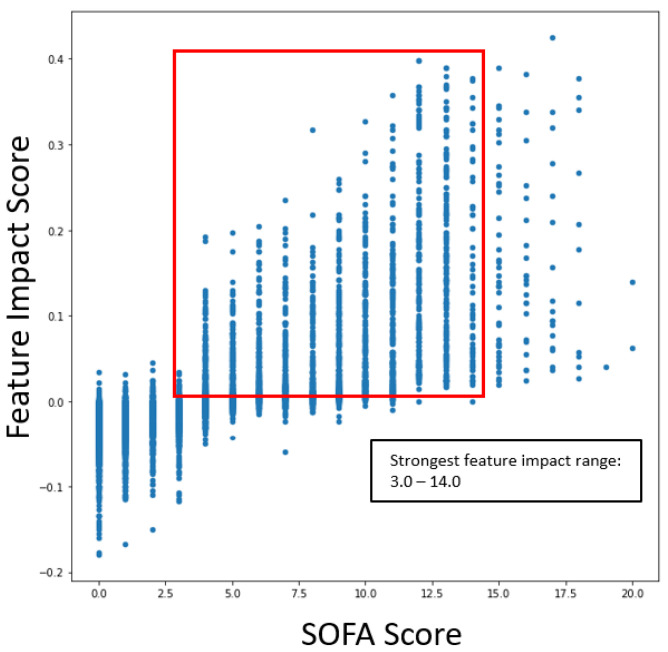
Range of impactful and validated biomarker values for COVID-19 survival prediction.

**Figure 10 ijms-24-06250-f010:**
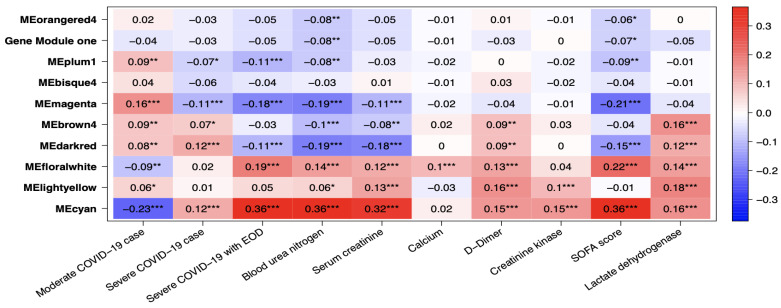
Heatmaps of module traits for severity and clinical biomarkers. The boxes indicate the correlation based on module eigengenes in the rows and traits in the column. The color legend—blue (negative correlation) and red (positive correlation). *p*-values represented by asterisks indicated significance. Asterix signs indicate the order of significance where ***—very significant, **—significant, and *—significant.

**Figure 11 ijms-24-06250-f011:**
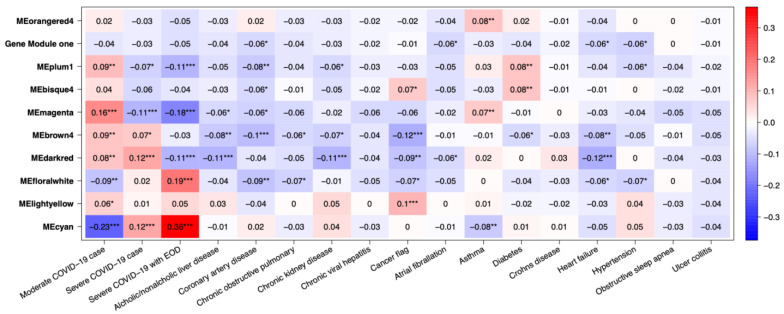
Heatmaps of module traits for severity and disease comorbidity. The boxes indicate the correlation based on module eigengenes in the rows and traits in the column. The color legend—blue (negative correlation) and red (positive correlation). *p*-values represented by asterisks indicated significance. Asterix signs indicate the order of significance where ***—very significant, **—significant, and *—significant.

**Figure 12 ijms-24-06250-f012:**
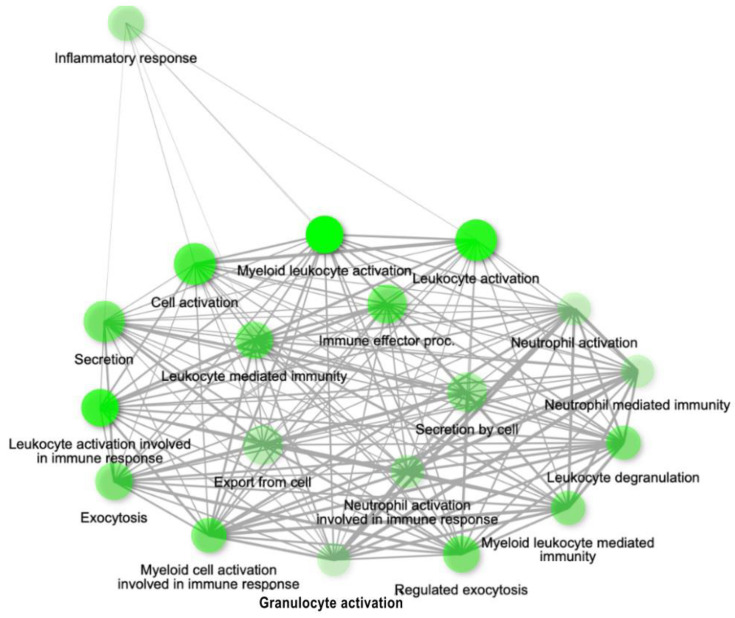
Gene network for top 20 enriched pathways for genes module, MEcyan based on GO biological process annotation.

**Figure 13 ijms-24-06250-f013:**
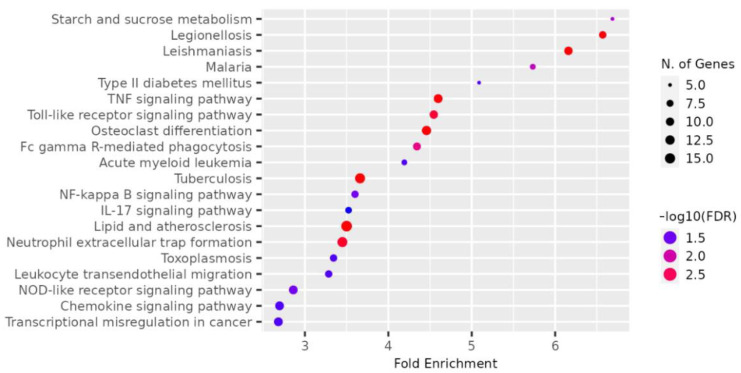
Dot plot for top 20 enriched pathways based for gene module, MEcyan based on KEGG.

**Figure 14 ijms-24-06250-f014:**
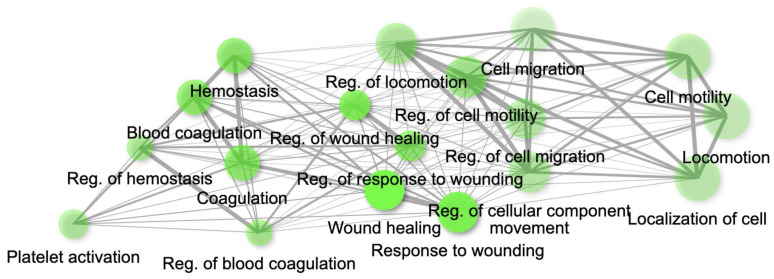
Gene network for top 20 enriched pathways for genes module, MEdarkred based on GO biological process annotation.

**Table 1 ijms-24-06250-t001:** Evaluation metrics for COVID-19 severity and survival models.

	Severity	Survival
Balanced accuracy	91.6%	99.1%
ROC-AUC	98.1%	99.9%

**Table 2 ijms-24-06250-t002:** Evaluation metrics for COVID-19 severity and survival models without comorbidities in the training data.

	Severity	Survival
Balanced accuracy	85.4%	69.8%
ROC-AUC	93.5%	87.8%

## Data Availability

All data used in the study were received from publicly available sources; the list of sources with the references is provided in the Appendix A.

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
