# Peer review of "Integrating AI/ML Models for Patient Stratification Leveraging Omics Dataset and Clinical Biomarkers from COVID-19 Patients: A Promising Approach to Personalized Medicine"

_ijms, 2023, doi:10.3390/ijms24076250_

Round 1
Reviewer 1 Report
In this study, the authors proposed a prediction model for Patient Stratification Leveraging Omics Dataset and Clinical Biomarkers from COVID-19 Patients. Although the idea is of interest, some major points should be addressed as follows:
1. The authors mentioned that they used different machine learning and deep learning models, but the results did not show clearly the performance among them. At least they must show some baseline comparisons to claim the significant choice.
2. Uncertainties of models should be reported.
3. The authors should have external validation data to evaluate the performance of models on unseen data.
4. The authors should compare the predictive performance to previously published works on the same problem/data.
5. In the SHAP analysis, why did the authors only list 20 features rather than other numbers? Did the authors conduct feature selection in this study?
6. Machine learning is well-known and has been used in previous biomedical studies i.e., PMID: 35832629, PMID: 33036150. Therefore, the authors are suggested to refer to more works in this description to attract a broader readership.
7. Besides SHAP, the authors should perform LIME analysis for model interpretation.
8. Quality of figures should be improved.
9. The use of AI and ML is confusing. ML is an AI model and therefore, it is better to use ML only.
Author Response
Reviewer 1
- The authors mentioned that they used different machine learning and deep learning models, but the results did not show clearly the performance among them. At least they must show some baseline comparisons to claim the significant choice.
The following updates were made in the manuscript:
- Part of sentence “and recurrent neural network variants such as and LSTM (Long Short-Term Memory) models [Hochreiter et al., 1997]” was removed from section 2.4.
- Several sentences: “Performance metrics for various modeling architectures were collected in Supplementary Tables (#,#). Missing values within the data was imputed using Multiple Iteration Chain Estimation (MICE) regression techniques (Azur et al., 2011), where each missing data point is iteratively learned from other available features. Boosted Decision Tree architectures, such as LightGBM, XGBoost, and CatBoost, have built in techniques for handling missing values [Khosravi et al. 2020], so evaluation metrics were reported twice (without MICE, with MICE) for each of these architectures.” were added to section 3.1
- Two new references were added: Azur et al., 2011; Khosravi, Pasha, et al 2020.
- Two more tables with quick description were added to supplementary data:
Table 11: Comparison of Balanced Accuracy and ROC-AUC among various modeling architectures for predicting COVID-19 severity. Boosted Decision Tree models, such as XGBoost, LightGBM, and CatBoost, all have built in methods for handling missing data, so evaluation metrics for these modeling architectures were reported with no data transformations and with MICE imputation. All other modeling architectures required missing data imputation, so MICE imputation was used.
|
Balanced Accuracy |
ROC-AUC |
|
|
LightGBM |
0.916 |
0.981 |
|
XGBoost |
0.925 |
0.978 |
|
CatBoost |
0.918 |
0.975 |
|
XGBoost + MICE |
0.92 |
0.976 |
|
LightGBM + MICE |
0.915 |
0.973 |
|
CatBoost + MICE |
0.913 |
0.973 |
|
SVC + MICE |
0.858 |
0.944 |
|
RandomForest + MICE |
0.885 |
0.961 |
|
MLP + MICE |
0.627 |
0.629 |
Table 12: Comparison of Balanced Accuracy and ROC-AUC among various modeling architectures for predicting COVID-19 survival. Boosted Decision Tree models, such as XGBoost, LightGBM, and CatBoost, all have built in methods for handling missing data, so evaluation metrics for these modeling architectures were reported with no data transformations and with MICE imputation. All other modeling architectures required missing data imputation, so MICE imputation was used.
|
Balanced Accuracy |
ROC-AUC |
|
|
LightGBM |
0.995 |
0.999 |
|
XGBoost |
0.992 |
0.999 |
|
CatBoost |
0.992 |
0.998 |
|
XGBoost + MICE |
0.994 |
0.998 |
|
LightGBM + MICE |
0.992 |
0.998 |
|
CatBoost + MICE |
0.992 |
0.998 |
|
SVC + MICE |
0.807 |
0.923 |
|
RandomForest + MICE |
0.871 |
0.988 |
|
MLP + MICE |
0.553 |
0.584 |
- Uncertainties of models should be reported.
Uncertainties in the models include:
- Sparsity in lab biomarker data restricts complete multivariate analysis for model explainability, as there would be a possibility for even more granular feature observations if lab biomarker data was more complete. There is a possibility that certain biomarkers, such as Hemoglobin A1C and Interleukin 8, which had more than 90% missing values, may have contributed to a significant impact on model predictability.
- Lack of diversity among observed comorbidities within patients in the dataset may have limited the potential observation for certain comorbidities to contribute to COVID-19 severity and survival based on model conclusions.
- The authors should have external validation data to evaluate the performance of models on unseen data.
Test holdout set is the “external validation set”. However there is no standard evaluation validation set used for benchmarking. The following sentences were added to end of 3rd paragraph of section 2.4: “Mention that validation set was derived from cross-fold validation splitting techniques from training set to finalize hyperparameter optimization and feature selection.
Mention results provided were for test holdout set.
Since duplicates were dropped before the training and test random splits, the test set represented an unseen feature set as there was no overlap with the training set.”
- The authors should compare the predictive performance to previously published works on the same problem/data.
The predictive performance accuracy was compared to Raman et al., 2023 and Ikemura et al 2021, which predicted COVID-19 severity and mortality with SHAP value results, respectively, the dataset that was used in this current study contained significantly more biomarker based data then both external studies – which allowed for SHAP value analysis of models that utilized biomarker based features in order to understand which biomarkers were highly contributive to COVID-19 survival and severity.
Thos lines were added to the discussions section two new references were added to the reference list.
- In the SHAP analysis, why did the authors only list 20 features rather than other numbers? Did the authors conduct feature selection in this study?
We selected top 20 features by absolute SHAP value scores, which are the essentially most impactful features on the model. Therefore, the following sentence was added to the end of section 3.2.1.: “The top 20 features were determined by the highest absolute SHAP value mean, which shows the most impactful features contributing to either class (survival / mortality) (severe case / moderate case).”
- Machine learning is well-known and has been used in previous biomedical studies i.e., PMID: 35832629, PMID: 33036150. Therefore, the authors are suggested to refer to more works in this description to attract a broader readership.
The introduction section was updated with a few sentences and two references suggested by the reviewer:
AI-powered models have been recently used with success to simulate and predict drug-drug interactions in different patient groups, which aids in preventing potential issues during the initial stages of drug development in clinical trials [Vo et al., 2022].
AI technologies can identify the particular pathways responsible for the development of disease-related syndromes, thereby improving the precision of patient cohort classi-fication [Lam et al., 2020].
- Besides SHAP, the authors should perform LIME analysis for model interpretation.
This analysis is no needed for this work because LIME analysis is applied only for local explainability, while SHAP is used for both global and local explainability. See the table with clarification below.
- Quality of figures should be improved.
Figures will be submitted separately because quality gets worse when figures are imbedded into a text.
- The use of AI and ML is confusing. ML is an AI model and therefore, it is better to use ML only. AI represents the overall intelligent system (integrating ML models into a clinic), whereas ML was the main form of analysis used to train models
“AI” was removed throughout the whole manuscript with only “ML” remaining to avoid confusion.
Reviewer 2 Report
The manuscript by Bello et al., “Integrating AI/ML Models for Patient Stratification Leveraging Omics Dataset and Clinical Biomarkers from COVID-19 Patients: A Promising Approach to Personalized Medicine” briefly demonstrated the strategy for the importance of clinical biomarkers in identifying high-risk patients and predicting disease progression. Overall, the manuscript is well demonstrated can be acceptable for publication in IJMS.
Comments
1. Lines 28-42, the information can be updated (minor) on the more lethal and infectious issue of COVID-19 variants and their constant evaluation threats with the mechanism than the parent i.e., “COVID-19 variants: What's the concern?”, “Waves and variants of SARS-CoV-2: understanding the causes and effect of the COVID-19 catastrophe”, and “Emergence of novel omicron hybrid variants: BA (x), XE, XD, XF more~”.
2. Figure 1 can be more elaborated and should be in brief.
3. Please provide one additional illustration as a summary of the finding and perspectives for COVID-19.
4. Discussion can be improved (minor). The finding should be more quantitatively discussed to highlight the significance.
Author Response
Reviewer 2.
- Lines 28-42, the information can be updated (minor) on the more lethal and infectious issue of COVID-19 variants and their constant evaluation threats with the mechanism than the parent i.e., “COVID-19 variants: What's the concern?”, “Waves and variants of SARS-CoV-2: understanding the causes and effect of the COVID-19 catastrophe”, and “Emergence of novel omicron hybrid variants: BA (x), XE, XD, XF more~”.
Introduction and discussion sections were updated following the comments from Reviewer 1 and Reviewer 2.
Also, the following verbiage was added to the first paragraph of introduction section: “Various SARS-CoV-2 variants led to more concerns due to lowered vaccine efficacy and rapidly decreasing immunity. On the other hand, waves and variants of SARS-CoV-2 helped understanding the causes and effect of the COVID-19 catastrophe related to respiratory illness and multiorgan failure. Emergence of novel omicron hybrid variants such as BA (x), XE, XD, XF, etc. demonstrated the role of population variability in disease severity and outcome.”
- Figure 1 can be more elaborated and should be in brief.
Figure 1 was updated accordingly and submitted in a separate file for better quality.
- Please provide one additional illustration as a summary of the finding and perspectives for COVID-19.
We think illustration figure may not be good representative of our finding summary, however we have included additional diagrams (Figures 22 and 23) for functional annotation of enriched cellular components in the supplementary section. This is done to put perspectives how some of the significant gene transcripts such as AMAD9 metalloproteinase protein, when in high abundance in the blood, may server an indicator of tissue injury, and its involvement in extracellular matrix formation and host’s inflammatory response to the COVID-19 in patients. This perspective has added to the discussion and conclusion section of the manuscript
In addition, we added one more paragraph into Discussion section to clarify and summarize results obtained and their future prospectives:
“Furthermore, the set of significant genes represented in the MEdarkred module (Supplementary Table 10) across EOD, SOFA, BUN; and heart failure, alcohol liver disease and kidney disease comorbidity include gene markers such as AMAD9, KIF1B, SLC22A4, MAPK14, MAP2K6, SLC2A3, ILI7RA which have been associated with COVID-19 disease severity, tissue injury and sepsis. For instance increased expression of ADAM9 and others member of metalloprotease protein family, in the blood has been associated with tissue damage and identified as a potential drug target for COVID-19 [Rinchai et al, 2016; Carapito et al, 2022; Safi et al, 2022; Fan et al, 2022; Huang et al, 2023; Cui et al, 2022]. Rinchai et al particularly highlight the involve-ment of AMAD9 metalloprotease in cellular adhesion, protein shedding, various cell population and activity including cell-cell interaction and fibroblast activity - a key part of connective tissue and extracellular matrix (ECM) formation. This is consistent with our findings where the annotation of the MEdarkred gene module is shown to be enriched in cellular component (supplementary Figures 22, 23) activities such as cell surface, focal adhesion, cell cortex, anchoring junction, stress fiber, platelet activities, myofibrils and actin cytoskeleton. These networks of biological and cellular activities further emphasize the importance of better understanding of observed abnormalities in extracellular matrix (ECM), inflammatory response, activities of metalloproteinase proteins and other key significant gene biomarkers in COVID-19 patients.”
- Discussion can be improved (minor). The finding should be more quantitatively discussed to highlight the significance.
Discussion section was re-worked with three new paragraphs added and the remaining ones updated:
“Uncertainties in the models include:
Sparsity in lab biomarker data restricts complete multivariate analysis for model explainability, as there would be a possibility for even more granular feature observations if lab biomarker data was more complete. There is a possibility that certain biomarkers, such as Hemoglobin A1C and Interleukin 8, which had more than 90% missing values, may have contributed to a significant impact on model predictability.
Lack of diversity among observed comorbidities within patients in the dataset may have limited the potential observation for certain comorbidities to contribute to COVID-19 severity and survival based on model conclusions.”
In addition, ML Biomarker findings and literature comparison was made as follows:
“The most impactful SOFA score range for COVID-19 severity and mortality was 4-14 and 3-14 respectively. These ranges tie closely with observations seen in [Yang et al., 2021] which suggests that patients with SOFA scores greater than or equal to 5 have a higher risk of mortality. In regards to LDH, [Li et al., 2020] reports patients with severe COVID-19 have greater than 350 U/L for mean concentration, which is slightly less than the lower value of the most impactful median LDH concentration range of 404 and 439 for severity and mortality respectively. A larger ratio between BUN and Creatinine levels has shown to be an indicator for COVID-19 severity with optimal thresholds at 33.5 and 51.7 according to [Ok et al.,2021]. The most impactful range for both BUN and Serum Creatinine, according to SHAP analysis, suggests that the most impactful BUN/Creatinine ratio is between 21 and 355, which is a larger range window than the study suggests. This wide range is likely due to the sparsity of biomarker data and the independent contributions of each BUN and Serum Creatinine to the model predictions; however, the higher impact on severity and survival seen for higher BUN and lower Serum Creatinine values does suggest that a larger ratio would be an indicator for both use cases. Similarly, a larger ratio between BUN and Albumin levels has shown to be an indicator for COVID-19 severity with optimal thresholds greater than 3.9 according to [Küçükceran et al., 2021], and SHAP analysis suggests that the most impactful range of BUN/Albumin is between 7.5 and 93. Additional features to consider moving forward would be the actual BUN to Serum Creatinine and BUN to Albumin ratios as actual values within the data, coupled with the presence of more lab biomarker data.”

Round 2
Reviewer 1 Report
My previous comments have been addressed.
Reviewer 2 Report
accept